# Robot-Assisted Minimally Invasive Esophagectomy versus Open Esophagectomy for Esophageal Cancer: A Systematic Review and Meta-Analysis

**DOI:** 10.3390/cancers14133177

**Published:** 2022-06-29

**Authors:** Stepan M. Esagian, Ioannis A. Ziogas, Konstantinos Skarentzos, Ioannis Katsaros, Georgios Tsoulfas, Daniela Molena, Michalis V. Karamouzis, Ioannis Rouvelas, Magnus Nilsson, Dimitrios Schizas

**Affiliations:** 1Surgery Working Group, Society of Junior Doctors, 151-23 Athens, Greece; stepesag@sni.gr (S.M.E.); iaziogas@sni.gr (I.A.Z.); k.skarentzos@gmail.com (K.S.); ioankats@med.uoa.gr (I.K.); 2First Department of Surgery, National and Kapodistrian University of Athens, Laikon General Hospital, 115-27 Athens, Greece; 3First Department of Surgery, Aristotle University of Thessaloniki, 541-24 Thessaloniki, Greece; tsoulfasg@auth.gr; 4Thoracic Service, Department of Surgery, Memorial Sloan Kettering Cancer Center, New York, NY 10065, USA; molenad@mskcc.org; 5Molecular Oncology Unit, Department of Biological Chemistry, National and Kapodistrian University of Athens, 115-27 Athens, Greece; mkaramouz@med.uoa.gr; 6Division of Surgery, Department of Clinical Science, Intervention and Technology (CLINTEC), Karolinska Institutet, 141-86 Stockholm, Sweden; ioannis.rouvelas@ki.se (I.R.); magnus.nilsson@ki.se (M.N.); 7Department of Upper Abdominal Diseases, Karolinska University Hospital, 171-77 Stockholm, Sweden

**Keywords:** robot-assisted minimally invasive esophagectomy, RAMIE, robotic esophagectomy, minimally invasive esophagectomy, open esophagectomy

## Abstract

**Simple Summary:**

Robot-assisted minimally invasive esophagectomy (RAMIE) constitutes a newly developed surgical technique for the treatment of resectable esophageal cancer, aiming to further improve the high morbidity and mortality associated with open esophagectomy. We performed a systematic review of the literature and compared the outcomes of RAMIE and open esophagectomy. RAMIE is a safe and feasible procedure, resulting in decreased cardiopulmonary morbidity, wound infections, blood loss, and hospital stays compared to open esophagectomy.

**Abstract:**

Robot-assisted minimally invasive esophagectomy (RAMIE) was introduced as a further development of the conventional minimally invasive esophagectomy, aiming to further improve the high morbidity and mortality associated with open esophagectomy. We aimed to compare the outcomes between RAMIE and open esophagectomy, which remains a popular approach for resectable esophageal cancer. Ten studies meeting our inclusion criteria were identified, including five retrospective cohort, four prospective cohort, and one randomized controlled trial. RAMIE was associated with significantly lower rates of overall pulmonary complications (odds ratio (OR): 0.38, 95% confidence interval (CI): [0.26, 0.56]), pneumonia (OR: 0.39, 95% CI: [0.26, 0.57]), atrial fibrillation (OR: 0.53, 95% CI: [0.29, 0.98]), and wound infections (OR: 0.20, 95% CI: [0.07, 0.57]) and resulted in less blood loss (weighted mean difference (WMD): −187.08 mL, 95% CI: [−283.81, −90.35]) and shorter hospital stays (WMD: −9.22 days, 95% CI: [−14.39, −4.06]) but longer operative times (WMD: 69.45 min, 95% CI: [34.39, 104.42]). No other statistically significant difference was observed regarding surgical and short-term oncological outcomes. Similar findings were observed when comparing totally robotic procedures only to OE. RAMIE is a safe and feasible procedure, resulting in decreased cardiopulmonary morbidity, wound infections, blood loss, and shorter hospital stays compared to open esophagectomy.

## 1. Introduction

Esophagectomy is the mainstay of treatment for resectable esophageal cancer [1]. The technical optimization of the procedure over the years has led to improved short- and long-term outcomes [2]. Despite this progress, esophagectomy is still a highly invasive and technically complex operation with considerable morbidity and mortality [3]. This aspect remains problematic, as postoperative complications are strongly associated with worse long-term outcomes and a higher cost of care [4,5]. Minimally invasive esophagectomy (MIE) has emerged as an alternative to the conventional open esophagectomy (OE) and has become more broadly adopted after early case series demonstrated lower complication and mortality rates compared to OE [6,7]. The TIME (traditional invasive vs. minimally invasive esophagectomy) trial and several meta-analyses validated these preliminary findings and convincingly showed that MIE, at least in the context of selected high-volume centers, is superior to OE regarding short- and long-term outcomes [8,9,10]. However, evidence from population-based cohort studies is less clear and in some cases even suggests worse short-term outcomes after MIE [11].

Minimally invasive esophagectomy is often used as an umbrella term for many different approaches, including the conventional thoracoscopic/laparoscopic MIE, hybrid MIE, and robot-assisted minimally invasive esophagectomy (RAMIE) [12]. More specifically, RAMIE was introduced as a solution to the inherent limitations and technical difficulties of conventional MIE, such as the most commonly used two-dimensional operative field vision, the limited surgical instrument mobility, and the well-documented long learning curve [13,14]. Since its implementation into clinical practice, large case series have consistently shown that RAMIE is safe and feasible with favorable short- and long-term outcomes [15,16,17,18], while a recent meta-analysis suggested that RAMIE is equivalent to conventional MIE [19]. 

Although RAMIE and conventional MIE have both demonstrated promising outcomes, OE is still a commonly utilized approach in many centers for the management of resectable esophageal cancer [20]. The results of the ROBOT randomized controlled trial showed that RAMIE was superior to OE for numerous outcomes of interest [21]. However, no systematic comparison between RAMIE and OE has been conducted to date. Thus, the aim of the current study is to perform a systematic review and meta-analysis to compare the short-term surgical and oncological outcomes between RAMIE and OE in patients with esophageal cancer, as the proposed benefits of RAMIE may soon lead to a shift in the standard of care for patients with resectable disease.

## 2. Materials and Methods

### 2.1. Study Design and Inclusion/Exclusion Criteria

This systematic review and meta-analysis was performed according to the PRISMA (Preferred Reporting Items for Systematic Reviews and Meta-analyses) guidelines and in line with the protocol developed and agreed by all authors (Appendix A) [22]. Our systematic review was registered at Open science (https://osf.io/prereg/ (accessed on 3 May 2022)) with a registration number of 10.17605/OSF.IO/XEPHB. The study selection criteria were defined by applying the PICO (Population/Participants, Intervention, Comparison, and Outcome) framework:Participants: Patients of any race, age, or sex undergoing esophagectomy for esophageal cancer.Interventions: RAMIE and OE. Studies were included independent of the surgical approach (i.e., transthoracic (Ivor-Lewis or McKeown), or transhiatal). Procedures were identified as RAMIE if the robotic approach was utilized for the thoracic phase of the operation, irrespective of the abdominal phase approach. The procedures were further classified as a) totally robotic (TRAMIE) if the robotic approach was utilized for both the thoracic and the abdominal part of the operation and b) hybrid if the robotic approach was utilized for the thoracic part of the operation or both types of the aforementioned approaches were included in a single sample.Comparison: Studies were deemed eligible only if RAMIE was directly compared to OE.Outcomes: The primary surgical outcome measures were the rates of overall complications, overall pulmonary complications, anastomotic leakage, 30-day mortality, and 90-day mortality. The primary oncological outcomes were the number of total lymph nodes resected, the margin-negative resection (R0) rate, the overall survival, (OS) and the disease-free survival (DFS). The secondary outcome measures were total operative time, estimated blood loss (EBL), hospital length of stay (LOS), and length of intensive care unit (ICU) stay as well the rates of pneumonia, acute respiratory distress syndrome (ARDS), postoperative hemorrhage, chylothorax, recurrent laryngeal nerve (RLN) palsy, atrial fibrillation, and wound infection.

Original clinical studies, including both randomized trials and non-randomized prospective/retrospective comparative studies, published in English, reporting on both RAMIE and OE for the outcomes of interest, were deemed eligible for inclusion. The exclusion criteria for the present systematic review were: (i) non-English language articles, (ii) irrelevant articles, (iii) non-comparative studies (<2 study arms), (iv) studies not directly comparing RAMIE and OE for the outcomes of interest, (v) animal and in-vitro studies, (vi) case reports, (vii) narrative or systematic reviews and meta-analyses, (viii) editorials, letters to the editor, perspectives, comments, and errata that did not provide any primary patient data, and (ix) published abstracts with no published full text. No search filters were applied.

All eligible studies were assessed for overlap based on the author list, center and country of origin, and dates of patient enrollment. Between studies with overlapping populations, we included those with the largest number of patients or reporting granular data on the outcomes of interest. On one occasion, analyses on additional outcomes were presented in two eligible articles; hence, data were extracted from both, but their population was not summed in the overall cohort numbers, as they constituted additional analyses on the same subjects [23,24]. Data from national registry studies were qualitatively assessed separately and were excluded from the final data synthesis to avoid overlap with single-center studies from the same country.

### 2.2. Literature Search Strategy

Eligible studies were identified by searching through the MEDLINE (via PubMed), Scopus, and Cochrane Library databases (end-of-search date: 2 May 2020) by two independent reviewers. The following search algorithm was used: ((robot-assisted OR robotic OR RAMIE OR minimally invasive) OR (transthoracic or open)) AND (oesophagectomy or esophagectomy). Any disagreements on article inclusion were resolved by a third reviewer when necessary. The reference lists of the included studies were also thoroughly searched to identify eligible, missed studies based on the “snowball” methodology [25].

### 2.3. Data Tabulation and Extraction

Data tabulation and extraction from the eligible records were performed using a standardized, pre-piloted form. Two independent reviewers extracted the data, and any disagreements were identified and resolved after the involvement of a third reviewer when necessary. Whenever a study provided data for multiple esophagectomy approaches, including those that were not of interest to this study (i.e., conventional MIE), we only extracted data pertaining to OE and RAMIE. The following data were collected: (i) study characteristics (first author, year of publication, study design, study center, study period for each intervention, and number of patients for each study group), (ii) patient characteristics (age in years, sex, body mass index (BMI) in kg/m^2^, prior comorbidities according to the Charlson–Deyo score or the American Society of Anesthesiologists (ASA) classification, clinical stage and tumor location, histology, degree of differentiation, and size in cm), (iii) primary surgical outcomes (overall complication rate, overall pulmonary complication rate, anastomotic leakage rate, 30-day mortality rate, and 90-day mortality rate), (iv) primary oncological outcomes (total number of lymph nodes resected and R0 resection rate), and (v) secondary outcomes (operative time in minutes, EBL in mL, ICU length of stay in days, LOS in days, pneumonia rate, ARDS rate, postoperative hemorrhage rate, chylothorax rate, RLN palsy rate, atrial fibrillation rate, and wound infection rate). Overall complication and overall pulmonary complication rates were defined as the number of patients who had at least one complication or one pulmonary complication, respectively. The umbrella term “overall pulmonary complications” included pneumonia, pneumothorax, pulmonary embolism, ARDS, and pleural effusion.

### 2.4. Quality of Evidence Assessment

An assessment of study quality for non-randomized studies was performed with the Newcastle–Ottawa scale (NOS) [26]; a score of ≥6 denoted high study quality. In the item assessing whether follow-up was long enough for outcomes to occur, the cut-off was a priori set at 90 days after esophagectomy, while regarding the item about follow-up adequacy, a rate of 90% was also adopted a priori.

For randomized controlled trials (RCTs), quality of evidence assessment was conducted with the Cochrane Collaboration’s tool, which assesses for selection, performance, detection, attrition, and reporting biases [27].

### 2.5. Statistical Analysis

#### 2.5.1. Data Pooling 

Categorical variables were summarized as frequencies and percentages, while continuous variables were summarized as means and standard deviations (SDs). When continuous data were presented as medians and ranges or interquartile ranges, we estimated the respective means and SDs by applying the methods described by Hozo et al. and Wan et al. [28,29]. We estimated all relative rates based on the available data for the variables of interest, and all data handling was performed according to the principles of the Cochrane Handbook [30]. 

#### 2.5.2. Meta-Analysis 

A meta-analysis was carried out to compare RAMIE and OE for all primary surgical, primary oncological, and secondary outcomes. Based on the extracted data, odds ratios (ORs) and 95% confidence intervals (CIs) were calculated by means of 2 × 2 tables for each categorical outcome; OR > 1 indicated that the outcome was more frequently present in the RAMIE group. A continuity correction of 0.5 in studies with zero cell frequencies was adopted [31,32]. Weighted mean differences (WMD) and 95% CIs were estimated for each continuous outcome; WMD > 0 corresponded to larger values in the RAMIE group. Between-study heterogeneity was assessed through the Cochran *Q* statistic and by estimating *I^2^*. High heterogeneity was confirmed with a significance level of *p* < 0.05 and *I^2^* ≥ 50%. Due to the significant between-study clinical heterogeneity, we used the random-effects model (DerSimonian–Laird) to calculate the pooled effect estimates for all outcomes [33]. An assessment of publication bias was conducted by means of funnel plots for each outcome of interest. Statistical significance was set at 0.05, and all *p*-values were two-tailed. Subgroup analyses were performed according to the characteristics of the RAMIE technique in order to compare the TRAMIE and hybrid approaches, as previously defined. Statistical analyses were performed using STATA IC 16.0 (StataCorp LLC, College Station, TX, USA).

## 3. Results

### 3.1. Study Selection and Characteristics

Through our systematic search, 3479 unique articles were retrieved, of which 111 underwent full-text evaluation for eligibility. Ultimately, 10 studies reporting on 1977 patients undergoing esophagectomy for esophageal cancer (674 RAMIE and 1303 open) fulfilled the inclusion criteria and were included in our quantitative data synthesis (Figure 1) [21,23,34,35,36,37,38,39,40,41]. Three analyses of the National Cancer Database (NCDB) (USA) were identified [42,43,44]; among these, we selected the one reporting on the largest number of patients and the pre-specified outcomes of interest to serve as a measure of comparison for the results of our systematic review [42]. Six of the included studies (60%) were published in 2019 or 2020. Detailed study and patient characteristics of the included studies are presented in Table 1 and Table 2, respectively.

### 3.2. Study Quality and Publication Bias Assessment

Of the 10 included studies in the final data synthesis, 9 were non-randomized studies (5 retrospective and 4 prospective), and 1 was a randomized controlled trial. The nine non-randomized studies were assessed using the NOS, with a mean score of 6.9 ± 1.45 (Appendix A).

One randomized controlled trial was included in the final analysis and was assessed separately using the Cochrane Collaboration’s tool. A low risk of bias was detected for selection, attrition, and reporting bias. Performance and detection bias could not be assessed, as blinding of the medical personnel performing the operations and assessing the outcomes is not possible in trials comparing surgical procedures (Appendix A).

The funnel plots assessing for publication bias are presented in Appendix A for the primary surgical and oncological outcomes and in Appendix A for the secondary outcomes. No publication was evident through visual assessment of the funnel plots.

### 3.3. Primary Surgical Outcomes

The results of the meta-analyses are summarized in Table 3.

#### 3.3.1. Overall Complication Rate

The overall complication rate was reported in five studies [24,34,36,37,41]. No statistically significant difference was found between the RAMIE group (27.88%; *n* = 109/391) and the OE group (33.93%; *n* = 303/893) (OR: 0.66, 95% CI: 0.42–1.05; *p* = 0.08). The statistical heterogeneity was high (*I^2^* = 50.17%). No statistically significant difference was detected between TRAMIE and OE (OR: 0.81, 95% CI: 0.49–1.34) (Figure 2A).

#### 3.3.2. Overall Pulmonary Complication Rate

The overall pulmonary complication rate was reported in six studies [21,35,37,38,40,41] and was significantly lower in the RAMIE group (14.29%; *n* = 49/343) compared to the OE group (25.32%; *n* = 174/687) (OR: 0.38, 95% CI: 0.26–0.56; *p* < 0.001). The statistical heterogeneity was low (*I^2^* = 0.00%). TRAMIE was also associated with a significantly lower overall pulmonary complication rate (OR: 0.38; 95% CI: 0.24–0.60) (Figure 2B).

#### 3.3.3. Anastomotic Leakage Rate

The anastomotic leakage rate was reported in 10 studies [21,23,34,35,36,37,38,39,40,41]. No statistically significant difference was found between the RAMIE group (6.82%; *n* = 46/674) and the OE group (6.06%; *n* = 79/1303) (OR: 0.93, 95% CI: 0.60–1.44; *p* = 0.76). The statistical heterogeneity was low (*I^2^* = 0.00%). No statistically significant difference was detected between TRAMIE and OE (OR: 0.91, 95% CI: 0.45–1.44) (Figure 2C). 

#### 3.3.4. Thirty-Day Mortality Rate

The thirty-day mortality rate was reported in three studies [21,23,35]. No statistically significant difference was found between the RAMIE group (0.81%; *n* = 2/248) and the OE group (1.49%; *n* = 6/402) (OR: 0.75, 95% CI: 0.15–3.78; *p* = 0.73). The statistical heterogeneity was low (*I^2^* = 0.00%) (Figure 2D). 

In comparison, the NCDB study by Weksler et al. reported a 30-day mortality rate of 5.6% for the RAMIE group and 2.7% for the OE group (*p* = 0.060) [42].

#### 3.3.5. Ninety-Day Mortality Rate

The ninety-day mortality rate was reported in six studies [21,24,34,35,37,41]. No statistically significant difference was found between the RAMIE group (2.59%; *n* = 10/421) and the OE group (2.239%; *n* = 18/808) (OR: 0.80, 95% CI: 0.31–2.05; *p* = 0.64). The statistical heterogeneity was low (*I^2^* = 0.00%). No statistically significant difference was detected between TRAMIE and OE (OR: 0.57, 95% CI: 0.17–1.86) (Figure 2E). 

In comparison, the NCDB study by Weksler et al. reported a 90-day mortality rate of 8.1% for the RAMIE group and 6.4% for the OE group (*p* = 0.399) [42].

### 3.4. Primary Oncological Outcomes

#### 3.4.1. Total Lymph Nodes Resected

The number of total lymph nodes resected was reported in eight studies [21,23,34,35,37,38,40,41]. No statistically significant difference was found between the RAMIE group (28.45 ± 14.71 lymph nodes resected) and the OE group (21.44 ± 15.53 lymph nodes resected) (WMD: 3.26 lymph nodes resected, 95% CI: −0.89–7.41; *p* = 0.12). The statistical heterogeneity was high (*I^2^* = 91.41%). No statistically significant difference was detected between TRAMIE and OE (WMD: 5.01 lymph nodes resected, 95% CI: −0.93–10.96) (Figure 3A). 

In comparison, the NCDB study by Weksler et al. reported that the mean number of resected lymph nodes was 16.25 ± 2.17 in the RAMIE group and 13.25 ± 2.17 in the OE group (*p* = 0.087) [42].

#### 3.4.2. R0 Resection Rate

The R0 resection rate was reported in six studies [21,23,34,35,37,39]. No statistically significant difference was found between the RAMIE group (98.02%; *n* = 495/505) and the OE group (95.52%; *n* = 1109/1161) (OR: 1.36, 95% CI 0.56–3.32; *p* = 0.85). The statistical heterogeneity was low (*I^2^* = 18.07%). No statistically significant difference was detected between TRAMIE and OE (OR: 1.20, 95% CI: 0.34–4.28) (Figure 3B).

In comparison, the NCDB study by Weksler et al. reported a R0 resection rate of 95.1% (*n* = 541/569) for the RAMIE group and 93.1% (*n* = 530/569) for the OE group (*p* = 0.165) [42].

#### 3.4.3. Overall Survival

The OS was reported by two studies [21,23]. Yun et al. reported a 1-year OS of 95.1% and 85.6% and a 3-year OS of 81.7% and 73.7% for the RAMIE and OE groups, respectively [23]. Van der Sluis et al. had a follow-up of 40 months, during which the median OS was not reached for either group [21].

In comparison, the NCDB study by Weksler et al. reported a median OS of 48 months (95% CI: 34–55 months) for the RAMIE group and 44 months (95% CI: 38–53 months) for the OE group (*p* = 0.53) [42].

#### 3.4.4. Disease-Free Survival

The DFS was reported by two studies [21,23]. Yun et al. reported a 1-year DFS of 54.4% and 53.2% and a 3-year OS of 49.2% and 45.6% for the RAMIE and OE groups, respectively [23]. Van der Sluis reported a median DFS of 26 months for the RAMIE group and 28 months for the OE group [21].

### 3.5. Secondary Outcomes

#### 3.5.1. Operative Time

The operative time was reported in eight studies [21,23,34,35,36,37,38,39] and was significantly longer in the RAMIE group (360.39 ± 115.44 min) compared to the OE group (306.21 ± 94.58 min) (WMD: 69.45 min, 95% CI: 34.39–104.42; *p* < 0.001). The statistical heterogeneity was high (*I^2^* = 96.58%). TRAMIE was also associated with significantly longer operative times (WMD: 69.50 min, 95% CI: 13.74–125.25) (Figure 4A).

#### 3.5.2. Estimated Blood Loss

Estimated blood loss was reported in eight studies [21,23,34,35,36,37,38,39] and was significantly lower in the RAMIE group (209.59 ± 169.02 mL) compared to the OE group (374.38 ± 415.34 mL) (WMD: −187.08 mL, 95% CI: −283.81–(−90.35); *p* < 0.001). The statistical heterogeneity was high (*I^2^* = 95.50%). TRAMIE was also associated with significantly lower estimated blood loss (WMD: −296.94 mL, 95% CI: −503.03–(−90.85)) (Figure 4B).

#### 3.5.3. ICU Length of Stay

The intensive care unit length of stay was reported in three studies [21,23,36]. No statistically significant difference was found between the RAMIE group (1.37 ± 0.56 days) and the OE group (1.59 ± 1.55 days) (WMD: −0.13 days, 95% CI: −0.28–0.02); *p* = 0.09). The statistical heterogeneity was low (*I^2^* = 27.67%) (Figure 4C).

#### 3.5.4. Length of Hospital Stay

The length of hospital stay was reported in nine studies [21,23,34,35,36,37,38,39,41] and was significantly shorter in the RAMIE group (17.10 ± 9.39 days) compared to the OE group (30.68 ± 23.88 days) (WMD: −9.22 days, 95% CI: −14.39–(−4.06); *p* < 0.001). The statistical heterogeneity was high (*I^2^* = 96.13%). TRAMIE was also associated with significantly a shorter length of hospital stay (WMD: −20.35, 95% CI: −34.75–(−5.95) (Figure 4D).

#### 3.5.5. Pneumonia Rate

The rate of pneumonia was reported in six studies [21,23,34,36,37,39] and was significantly lower in the RAMIE group (7.94%; *n* = 42/529) compared to the OE group (14.90%; *n* = 181/1146, 15.79%) (OR: 0.39, 95% CI: 0.26–0.57; *p* < 0.001). The statistical heterogeneity was low (*I^2^* = 0.00%). The rate of pneumonia was also significantly lower in the TRAMIE group compared to the OE group (OR: 0.43, 95% CI: 0.23–0.80) (Figure 5A). 

#### 3.5.6. ARDS Rate

The rate of ARDS was reported in three studies [21,23,41]. No statistically significant difference was found between the RAMIE group (0.46%; *n*= 1/217) and the OE group (1.34%; *n* = 4/298, 1.34%) (OR: 0.27, 95% CI: 0.04–1.73; *p* = 0.17). The statistical heterogeneity was low (*I^2^* = 0.00%) (Figure 5B).

#### 3.5.7. Atrial Fibrillation Rate

The rate of atrial fibrillation was reported in five studies [21,23,34,35,36] and was significantly lower in the RAMIE group (6.79%, *n* = 29/427) compared to the OE group (8.46%, *n* = 54/638) (OR: 0.53, 95% CI: 0.29–0.98; *p* = 0.04). The statistical heterogeneity was low (*I^2^* = 14.60%). No statistically significant difference was detected between TRAMIE and OE (OR: 1.08, 95% CI: 0.47–2.48) (Figure 5C).

#### 3.5.8. Postoperative Hemorrhage Rate

The rate of postoperative hemorrhage was reported in four studies [21,23,34,35]. No statistically significant difference was found between the RAMIE group (1.18%; *n*= 4/339) and the OE group (2.51%, *n* = 12/479) (OR: 0.59, 95% CI: 0.18, 1.92]; *p* = 0.38). The statistical heterogeneity was low (*I^2^* = 0.00%). No statistically significant difference was detected between TRAMIE and OE (OR: 0.83, 95% CI: 0.11–6.53) (Figure 6A).

#### 3.5.9. Chylothorax Rate

The rate of chylothorax was reported in seven studies [21,23,34,35,37,39,41]. No statistically significant difference was found between the RAMIE group (5.39%; *n* = 29/538) and the OE group (3.01%; *n* = 33/1095) (OR: 1.31, 95% CI: 0.75–2.29; *p* = 0.35). The statistical heterogeneity was low (*I^2^* = 0.00%). No statistically significant difference was detected between TRAMIE and OE (OR: 0.69, 95% CI: 0.21–2.31) (Figure 6B).

#### 3.5.10. Recurrent Laryngeal Nerve Palsy Rate

The rate of RLN palsy was reported in seven studies [21,23,34,35,36,38,39]. No statistically significant difference was found between the RAMIE group (13.99%; *n* = 67/479) and the OE group (10.41%; *n* = 84/807) (OR: 1.31, 95% CI 0.90–1.90; *p* = 0.16). The statistical heterogeneity was low (*I^2^* = 0.00%). No statistically significant difference was detected between TRAMIE and OE (OR: 1.46, 95% CI: 0.59–3.64) (Figure 6C).

#### 3.5.11. Wound Infection Rate

The rate of wound infection was reported in four studies [21,34,37,38] and was significantly lower in the RAMIE group (1.25%; *n* = 4/319) compared to the OE group (5.95%; *n* = 38/637) (OR: 0.20, 95% CI: 0.07–0.57; *p* < 0.001). The statistical heterogeneity was low (*I^2^* = 0.00%). TRAMIE was also associated with a significantly lower wound infection rate (OR: 0.21, 95% CI 0.04–0.96) (Figure 6D).

## 4. Discussion

Open esophagectomy, a highly invasive procedure associated with significant morbidity and mortality [1,3], is still a commonly utilized technique for the management of resectable esophageal cancer in many centers internationally [20]. Robot-assisted minimally invasive esophagectomy was introduced as a potential solution to these problems [13]. This systematic review and meta-analysis of 2046 patients, to our knowledge the first one directly comparing open to robotic MIE, robustly shows an advantage for robotic MIE with regard to several short-term outcomes.

The overall complication rate following esophagectomy is directly linked to the mortality rate of the procedure, as most postoperative deaths occur in patients with multiple complications [45]. No statistically significant difference was observed in the overall complication rate and 30-day and 90-day mortality rates, suggesting that RAMIE might be an overall safe procedure. The limited number of studies reporting on these outcomes prevent us from drawing any definitive conclusions in spite of the lower absolute rates seen with RAMIE. The length of hospital stay was much shorter with RAMIE. As with MIE, the less invasive nature of the procedure minimizes surgical trauma and seemingly allows for faster recovery [8]. However, postoperative complications also prolong hospitalization [46] and thus may be partially responsible for the longer LOS of the OE group, with a WMD of 9.22 days (95% CI: [−14.39, −4.06]). An interesting observation is that in our study the average LOS in both groups was prolonged compared to what has previously been reported in the literature regarding both open and robot-assisted approaches [15,47]. Again, this may be explained by the fact that some patients in both groups experienced severe complications that may have skewed the LOS towards higher average values. TRAMIE, in particular, was associated with about a 20-day shorter length of hospital stay, contrary to hybrid operations which did not statistically differ from the open ones, and, thus, was the main contributary to the observed effect. Future studies specifically focusing on the LOS in patients without any postoperative complications may provide more insight into the true effect of the technique itself on this outcome. Even though only a minority of studies focused on the quality of life and postoperative recovery, data from Van der Sluis et al. and Sugawara et al. suggest RAMIE may have an advantage over OE in regard to these outcomes that extends beyond the initial hospitalization period [21,40].

Pulmonary complications constitute a major cause of morbidity and account for a large proportion of postoperative mortality following esophagectomy [45,48]. Among all pulmonary complications, pneumonia is the most common and has been identified as an independent prognostic factor of mortality and worse long-term outcomes [3,49,50]. Soon after the introduction of conventional MIE, the decreased incidence of pulmonary complications stood out as its main advantage over the open approach [6,7] In the TIME randomized controlled trial, the MIE group had an almost 70% relative risk reduction of pneumonias the first 2 weeks after the procedure [8]. The lower incidence of pulmonary complications with MIE has been attributed to many different factors. Compared to the lateral decubitus position used during OE, the prone positioning often employed during MIE decreases the compression of the lung parenchyma and may bypass the limitation of single lung ventilation by avoiding total lung collapse [8,51]. A significant proportion of the included studies in this meta-analysis used the prone [38,41] or semi-prone [21,23,34] positioning during the thoracic part of the robot-assisted procedure. In addition, the minimally invasive nature of the procedure decreases pulmonary tissue trauma and postoperative pain, leading to decreased inflammation, more effective breathing mechanics, and decreased basal atelectasis [52,53]. Van der Sluis et al. found that postoperative pain was significantly lower with RAMIE compared to the open procedure [21]. As with the conventional MIE, this can be attributed to the avoidance of thoracotomy [8]. Robot-assisted surgery also has the advantage of three-dimensional operative field visualization and offers seven degrees of freedom using articulated instruments [15]. This may enable improved preservation of the pulmonary branches of the vagus nerve, which regulates many important pulmonary functions and reflexes and, if compromised, may set the stage for postoperative pulmonary complications to occur [34].

Anastomotic leakage is common but is also one of the most feared complications following esophagectomy. It is associated with significant morbidity, as it tends to co-occur with other complications, prolonged hospitalization, and high mortality [3,54]. In terms of technical considerations, leakage is more common with cervical anastomoses and the McKeown approach compared to intrathoracic anastomoses and the Ivor-Lewis approach [54,55]. In our meta-analysis, the rate of anastomotic leakage was almost identical between RAMIE and OE. As with conventional MIE, RAMIE does not seem to offer any benefit for this outcome per se but appears to be as safe as OE nonetheless [56]. Another complication that is often associated with postoperative morbidity following esophagectomy is atrial fibrillation [57,58]. We found that atrial fibrillation was observed at significantly higher rates in the OE. The pathophysiology of atrial fibrillation after esophagectomy is complex and involves multiple factors. In this instance, the decreased rate in RAMIE might be explained by a combination of decreased intravascular depletion through less EBL and decreased oxidative stress because of fewer infectious complications and better lung ventilation in the prone position [57]. Interestingly TRAMIE was not associated with lower rates of atrial fibrillation, while hybrid procedures were found to lead to significantly lower rates, but the limited number of included studies could lead to this result.

Recurrent laryngeal nerve palsy and chylothorax rates were slightly higher in the RAMIE group. The relatively high RLN palsy rates observed after RAMIE in the past were mainly attributed to the extensive en bloc lymphadenectomy of the superior mediastinum [59]. The findings of Gong et al. support this association, as higher numbers of resected superior mediastinal lymph nodes were also combined with a higher incidence of vocal cord paralysis in the RAMIE group [34]. Van der Sluis et al. also attributed the relatively high incidence of chylothorax to the same phenomenon [21]. Regardless of that, the differences in the rates of RLN palsy and chylothorax between the two groups were found to be non-statistically significant.

RAMIE displayed a clear advantage over OE in terms of EBL. In line with MIE, the minimally invasive nature of RAMIE reduces tissue trauma and thus decreases the risk of blood vessel injury. The prone positioning of the patient during MIE and RAMIE allows for better visualization of the operative field compared to the lateral decubitus position of the open approach. In the latter case, blood tends to pool within the operative field and consequently obstructs the intraoperative view [60]. As a result, the prone position allows for higher surgical precision in both dissection and hemostasis, hence minimizing blood loss [61]. RAMIE has the additional advantage of three-dimensional visualization of the operating field as well as greater mobility and precision of the surgical instrumentation. This may explain why RAMIE results in lower EBL compared to both OE and MIE [19,38,62]. This decrease in intraoperative blood loss may also lead to reduced transfusion requirements, which are generally associated with inferior outcomes [63]. Nevertheless, the main drawback of RAMIE was the significantly longer operative time compared to the open esophagectomy group [59,64]. The docking and undocking of the robotic equipment is a lengthy process that prolongs the total operative time [64]. RAMIE has a learning curve of around 70 cases, but as experience accumulates, intraoperative parameters, including operative time and EBL are expected to decrease [65].

Open esophagectomy and RAMIE were comparable in terms of oncological outcomes and resulted in similar numbers of resected lymph nodes and R0 resection rates. Both these outcomes are strongly linked to long-term survival following esophagectomy [66,67,68]. Compared to the respective groups from the national database analysis by Weksler et al., the pooled numbers of resected lymph nodes in the current meta-analysis in both the RAMIE and OE groups were higher, while the R0 resection rates were similar [42]. Van der Sluis et al. reported comparable OS and DFS rates up to 5 years between the two groups [21]. Weksler et al. also came to the same conclusion about overall survival up to 4 years [42]. In comparison, Yun et al. found that RAMIE had an advantage in 1-year (95.1% in RAMIE vs. 85.6% in OE) and 3-year OS (81.7% in RAMIE vs 73.7% in OE) [23]. Although these preliminary data suggest that RAMIE demonstrates equivalent or even superior long-term survival compared to OE, additional data are required to deduce more meaningful conclusions.

Nonetheless, certain limitations should be considered when interpreting the results of the present study. First, we included patients undergoing esophagectomy with either the transthoracic or the transhiatal approach. Although both approaches lead to similar outcomes, the transhiatal approach offers a limited capacity for an extended lymphadenectomy [47]. Similar concerns about the heterogeneity of our sample arise due to varying definitions and assessment protocols for complications, as well as technical variations, such as the role of the robot in creating the anastomosis and performing the lymphadenectomy, which warrant a careful interpretation of our results. However, performing a subgroup analysis for any of these parameters was not feasible, as the available data were limited and in some occasions the approaches differed between the RAMIE and OE groups within the same study [37,39,40,41]. Second, a very small proportion of patients with high-grade dysplasia/esophageal cancer in situ originating from two studies [35,37] had to be included in the patient sample, as selective data extraction for esophageal cancer patients only was not feasible; we decided to include these two studies as the significant increase in the study sample and the accuracy of the effect estimates was deemed to be more important than the slight increase in disease heterogeneity. In addition, the RAMIE group included a slightly higher proportion of stage I patients and a lower proportion of stage III patients compared to the OE group. Third, none of the included studies evaluated the health-related economic cost difference between the two procedures. Although the robotic approaches in surgery are typically associated with higher costs compared to their open counterparts [64], future studies focusing on the cost effectiveness of the different approaches are warranted. Fourth, RAMIE is a relatively new procedure compared to OE, and surgical expertise varied among the included studies; however, the effects of the learning curve on the outcomes could not be sufficiently evaluated. It is therefore expected that as surgeons acquire more expertise in RAMIE, the outcomes will continue to improve [65,69]. Finally, the limited study sample for certain outcomes of interest deserves careful interpretation of the non-statistically significant differences and also precluded us from conducting meta-analyses on the outcomes of OS and DFS.

## 5. Conclusions

In conclusion, the results of this systematic review and meta-analysis suggest that RAMIE is a feasible and safe procedure in carefully selected patients. Compared to OE, it may lead to less EBL, shorter LOS, and reduced rates of pulmonary morbidity, atrial fibrillation, and wound infection. More studies in the form of randomized controlled trials and well-designed large population-based studies are required to evaluate the potential advantage of RAMIE in terms of overall complications and long-term oncological results as well as the impact of the different surgical approaches (transthoracic and transhiatal) on outcomes.

## Figures and Tables

**Figure 1 cancers-14-03177-f001:**
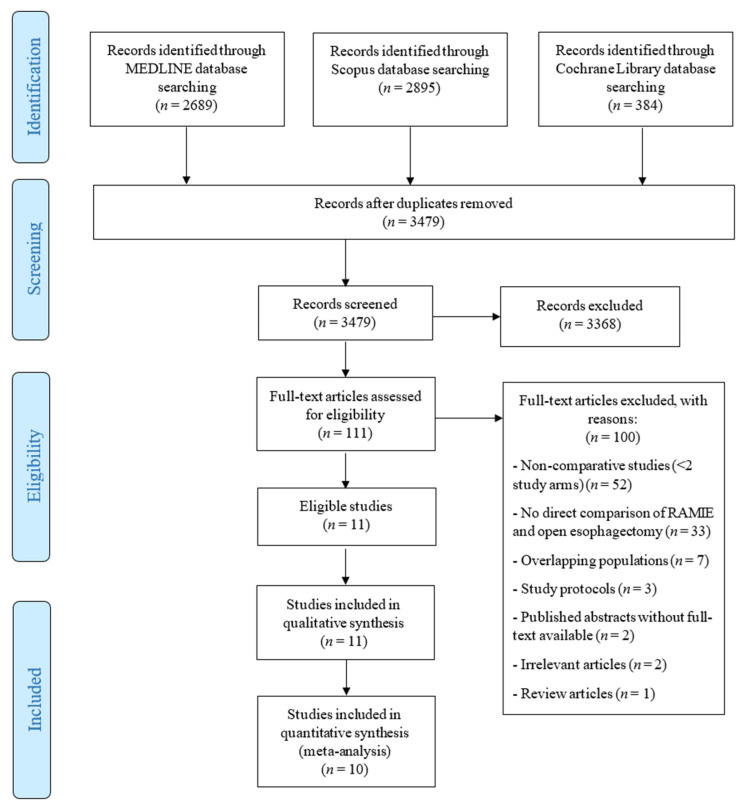
Preferred Reporting Items for Systematic Reviews and Meta-Analyses (PRISMA) flow diagram of the study selection process.

**Figure 2 cancers-14-03177-f002:**
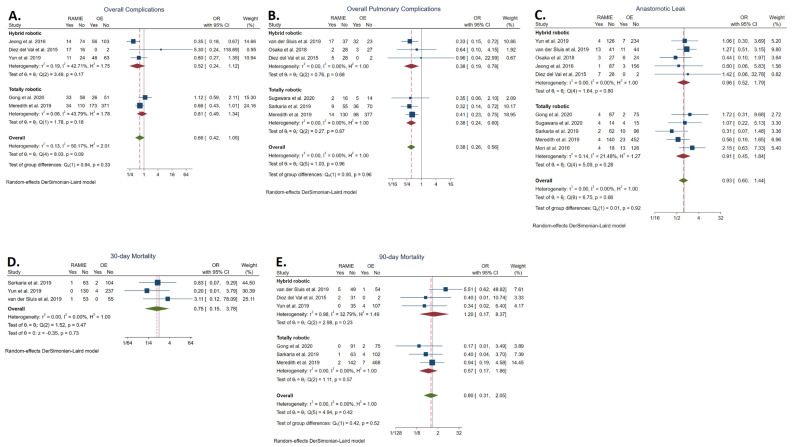
Forest plots of overall complication rate (**A**), overall pulmonary complication rate (**B**), anastomotic leakage rate (**C**), 30-day mortality rate (**D**), and 90-day mortality rate (**E**) [21,23,24,25,34,35,36,37,38,39,40,41].

**Figure 3 cancers-14-03177-f003:**
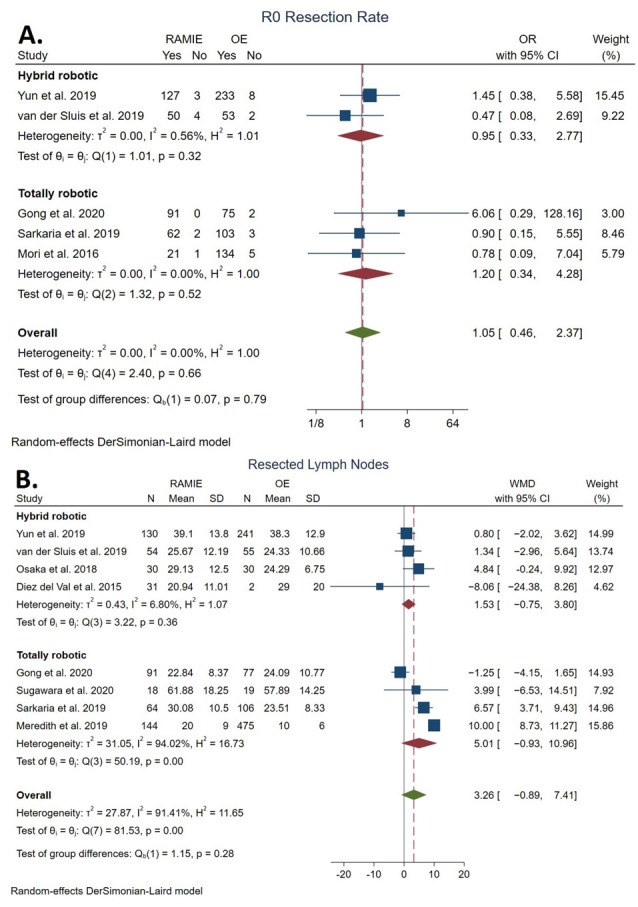
Forest plots of total lymph nodes resected (**A**) and margin-negative resection (R0) rate (**B**) [21,23,34,35,37,38,39,40,41].

**Figure 4 cancers-14-03177-f004:**
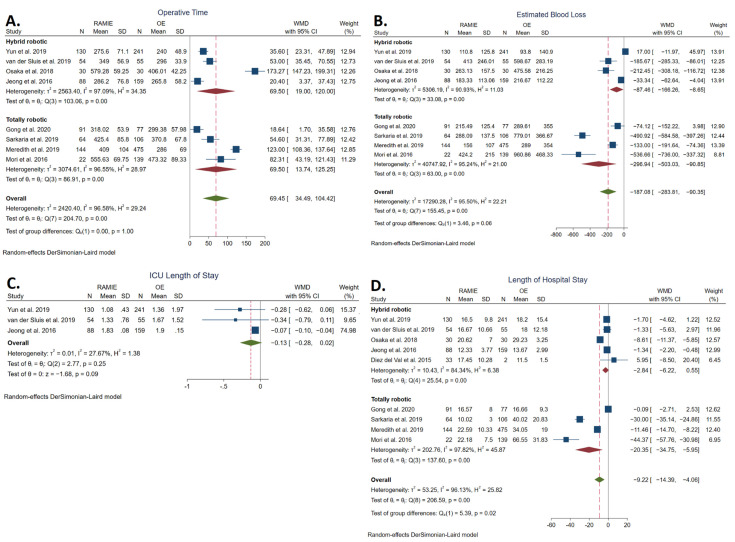
Forest plots of operative time (**A**), estimated blood loss (**B**), intensive care unit (ICU) length of stay (**C**), and hospital length of stay (**D**) [21,23,34,35,37,38,39,40,41].

**Figure 5 cancers-14-03177-f005:**
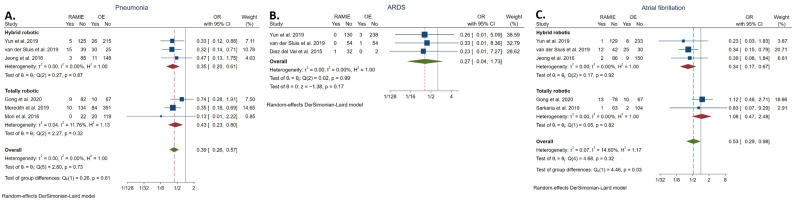
Forest plots of pneumonia rate (**A**), acute respiratory distress syndrome rate (**B**), and atrial fibrillation rate (**C**) [21,23,34,35,36,37,39,41].

**Figure 6 cancers-14-03177-f006:**
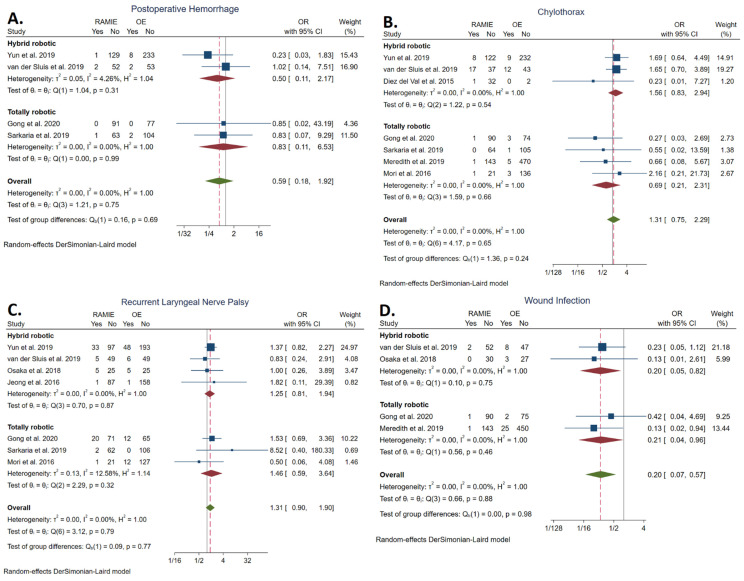
Forest plots of postoperative hemorrhage rate (**A**), chylothorax rate (**B**), recurrent laryngeal nerve palsy rate (**C**), and wound infection rate (**D**) [21,23,34,35,36,37,38,39,41].

**Table 1 cancers-14-03177-t001:** Characteristics of the included studies.

**Single-Center Studies**							
**Author**	**Year**	**Country**	**Study Period**	**Type of Study**	**RAMIE**	**OE**	**RAMIE Technique**
Gong et al.	2020	China	Jan 2016 to Dec 2018	Retrospective cohort	77	91	Robotic thoracic + abdominal phase; cervical anastomosis
Sugawara et al.	2020	Japan	Apr 2015 to Jan 2017	Prospective cohort	18	19	Robotic transhiatal combined with a video-assisted cervical approach; cervical anastomosis
Sarkaria et al.	2019	USA	Mar 2012 to Aug 2014	Prospective cohort	64	106	Robotic thoracic + abdominal phase; cervical or intrathoracic anastomosis
Yun et al.	2019	South Korea	Jan 2012 to Dec 2016	Retrospective cohort	130	241	Robotic thoracic phase; robotic abdominal or laparoscopic phase; cervical or intrathoracic anastomosis
Meredith et al.	2019	USA	1999 to 2016	Retrospective cohort	144	475	Robotic thoracic + abdominal phase; intrathoracic anastomosis
van der Sluis et al.	2019	The Netherlands	Jan 2012 to Aug 2016	Randomized controlled trial	54	55	Robotic thoracic phase; laparoscopic abdominal phase; cervical anastomosis
Osaka et al.	2018	Japan	Jun 2010 to Dec 2013 (RAMIE), 2006–2010 (OE)	Retrospective cohort	30	30	Robotic thoracic phase
Jeong et al.	2016	South Korea	Dec 2012 to Apr 2015	Retrospective cohort	88	159	Robotic thoracic phase; open abdominal phase; cervical anastomosis
Mori et al.	2016	Japan	Nov 2012 to Jul 2014 (RAMIE), May 2008 to Jul 2014 (OE)	Prospective cohort	22	139	Robotic transhiatal combined with a video-assisted cervical approach; cervical anastomosis
Diez del Val et al.	2015	Spain	Dec 2009 to N/A	Prospective cohort	33	2	Robotic thoracic or transhiatal phase; intrathoracic or cervical anastomosis (respectively)
**Total**			1999 to 2019		674	1303	
**National Registry Studies**							
**Author**	**Year**	**Country**	**Study Period**	**Type of Study**	**RAMIE**	**Open**	
Weksler et al.	2017	USA	2010 to 2013	Retrospective cohort	569	569	N/A

N/A: not available; RAMIE: Robot-assisted minimally invasive esophagectomy; OE: Open esophagectomy.

**Table 2 cancers-14-03177-t002:** Demographic, preoperative, and tumor characteristics.

		RAMIE (*n* = 674)	OE (*n* = 1303)
Demographic characteristics			
Age (years)		63.5 ± 8.5	62.7 ± 9.27
Male/Female		545 (85.0)/96 (15.0)	1133 (87.1)/168 (12.9))
BMI (kg/m^2^)		25.8 ± 6.8	25.7 ± 5.4
Preoperative characteristics			
ASA physical status			
	1–2	195 (55.7)	427 (60.7)
	3–4	155 (44.3)	276 (39.3)
Charlson–Deyo score			
	0	18 (16.5)	17 (17.7)
	1	31 (28.4)	33 (34.4)
	2	42 (38.5)	35 (36.5)
	3	14 (12.8)	11 (11.5)
	4	4 (3.7)	0 (0.0)
Tumor characteristics			
Tumor location			
	Proximal 1/3	57 (13.9)	94 (14.1)
	Middle 1/3	136 (33.3)	246 (36.9)
	Distal 1/3 + GEJ	216 (52.8)	327 (49.0)
Tumor histology			
	Squamous cell carcinoma	271 (72.7)	472 (74.4)
	Adenocarcinoma	100 (26.8)	154 (24.3)
	Other	2 (0.5)	8 (1.3)
Tumor differentiation			
	G1 (well-differentiated)	13 (14.3)	11 (14.3)
	G2 (moderately differentiated)	54 (59.3)	51 (66.2)
	G3 (poorly differentiated)	24 (26.3)	15 (19.5)
Clinical stage (according to TNM)			
	Stage 0	1 (0.2)	2 (0.2)
	Stage I	243 (39.9)	318(30.3)
	Stage II	181 (29.7)	333 (31.7)
	Stage III	178 (29.2)	375 (35.7)
	Stage IV	6 (1.0)	23 (2.2)
Tumor size (cm)		5.0 ± 2.1	4.4 ± 1.8
Neoadjuvant treatment		292 (51.8)	596 (59.3)

Values are given as means ± SD or *n* (%). ASA = American Society of Anesthesiologists; GEJ = gastroesophageal junction; RAMIE: Robot-assisted minimally invasive esophagectomy; OE: Open esophagectomy.

**Table 3 cancers-14-03177-t003:** Summary of meta-analyses for all outcomes.

Outcomes		Sum RAMIE	Sum Open	OR/WMD	95% CI	*p* Value	*I^2^* (%)	Result
Primary surgical								
	Overall complications *	109/391	303/893	0.66	[0.42, 1.05]	0.08	50.17	NS
	Overall pulmonary complications *	49/343	174/687	0.38	[0.26, 0.56]	<0.001	0.00	Favors RAMIE
	Anastomotic leakage	46/674	79/1303	0.93	[0.60, 1.44]	0.76	0.00	NS
	30-day mortality	2/248	6/402	0.75	[0.15, 3.78]	0.73	0.00	NS
	90-day mortality	10/421	18/808	0.80	[0.31, 2.05]	0.64	0.00	NS
Primary oncological								
	Lymph nodes resected	28.45 ± 14.71	21.44 ± 15.53	3.26	[−0.89, 7.41]	0.12	91.41	NS
	R0 resection	495/649	1047/1092	1.34	[0.56, 3.17]	0.33	13.82	NS
Secondary								
	Operative time ^†^	360.39 ± 115.44	306.21 ± 94.58	69.45	[34.39, 104.42]	< 0.001	96.58	Favors OE
	EBL ^‡^	209.59 ± 169.02	374.38 ± 415.34	−187.08	[−283.81, −90.35]	< 0.001	95.50	Favors RAMIE
	ICU length of stay ^§^	1.37 ± 0.56	1.59 ± 1.55	−0.13	[−0.28, 0.02]	0.09	27.67	NS
	LOS ^§^	17.10 ± 9.39	30.68 ± 23.88	−9.22	[−14.39, −4.06]	<0.001	96.13	Favors RAMIE
	Pneumonia	42/529	181/1146	0.39	[0.26, 0.57]	<0.001	0.00	Favors RAMIE
	ARDS	1/217	4/298	0.27	[0.04, 1.73]	0.17	0.00	NS
	Atrial fibrillation	29/427	54/638	0.53	[0.29, 0.98]	0.04	14.60	Favors RAMIE
	Postoperative hemorrhage	4/339	12/479	0.59	[0.18, 1.92]	0.38	0.00	NS
	Chylothorax	29/538	33/1095	1.31	[0.75, 2.29]	0.35	0.00	NS
	RLN palsy	67/479	84/807	1.31	[0.90, 1.90]	0.16	0.00	NS
	Wound infection	4/319	38/637	0.20	[0.07, 0.57]	<0.001	0.00	Favors RAMIE

* expressed as patients affected/total patients, ^†^ in minutes; ^‡^ in mL; ^§^ in days; RAMIE: robot-assisted minimally invasive esophagectomy; OE: open esophagectomy; OR: odds ratio; WMD: weighted mean difference; CI: confidence interval; EBL: estimated blood loss; ICU: intensive care unit; LOS: length of hospital stay; ARDS: acute respiratory distress syndrome; RLN: recurrent laryngeal nerve.

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
