# Peer review of "Robot-Assisted Minimally Invasive Esophagectomy versus Open Esophagectomy for Esophageal Cancer: A Systematic Review and Meta-Analysis"

_cancers, 2022, doi:10.3390/cancers14133177_

Round 1
Reviewer 1 Report
First I want to thank for the given opportunity to review the manuscript titled "Robotic-Assisted Minimally Invasive Esophagectomy Versus Open Esophagectomy for Esophageal Cancer: A Systematic Review and Meta-Analysis" submitted by Esagian and colleagues.
As I am a general thoracic surgeon experienced in minimally invasive and open esophagectomy I really appreciated this systematic review due to its necessity. In my opinion this excellently written manuscript covers a complex and complicated field of surgery which emerges increasing interest in daily surgical practice.
Major queries: none.
Minor queries:
1) page 10; paragraph 3.3.5:
I suppose this should mean Ninety-day mortality rate instead of "Thirty-day....".
2) page 18; paragraph 3.5.9:
I suppose this should mean postoperative chylothorax rate instead of "postoperative hemorrhage ....".
Author Response
The authors would like to thank the reviewer for their comments. We have revised the titles of both paragraphs, as correctly pointed out in the minor queries section of the review, to appropriately reflect the content of each paragraph.
Reviewer 2 Report
In this paper, I could find many errors in the paper selection and the data. Meta-analysis is not just gathering papers with same title. The authors should check whether patients’ group and values of the data are comparable. The incorrect data collection can lead to incorrect conclusion.
1). Yun’s paper was published in 2020 not in 2019. In Yun’s paper, overall complication was not reported, and 90-day mortality rate was not also reported.
2) In Sharkaria’s paper, the leakage rate is more than grade 2. Therefore it is not comparable to other papers. The number of harvested lymph node is different from original paper. In Sharkaria’s paper most values about hospital stays and ICU stays was suggested as median value. How the authors could get the mean value used in the manuscript?
3) In Jeong’s paper, blood loss amount is different from original paper. The authors used median value of hospital stay in Jeong’s paper. It is not possible to use median value in meta-analysis comparison.
4) In Meredith’s paper, the open surgery was sum of transthoracic and transhiatal surgery. Those two surgery is totally different surgery. Therefore transhiatal esophagectomy should be removed in the analysis. In Meredith’s paper, they did not suggested whether the mortality rate was 30-day or 90-day mortality. However, could you know it was 90-day mortality? The authors also used mean value from Meredith’s paper, However, several outcomes were presented as median value in original paper.
5) In Osaka’s paper, the number of harvested lymph nodes just included the number of mediastinal lymph nodes, not whole lymph nodes. It is not comparable to other papers. Operation time and blood loss amount is different from original paper.
6) Mori and Sugawara’s paper come from same center and the data is shared in both paper. Furthermore it is about the non-transthoracic cervical robotic esophagectomy. Technically it is totally different operation to other papers.
7) Diez del Val paper includes only 2 cases of open esophagectomy. How can the paper be used for comparative study only just including 2 cases?
Author Response
Thank you for your comments.
1. As you can see in the website of the Journal, the article by Yun et al. was originally published in 2019 (https://academic.oup.com/dote/article/33/5/doz071/5610078), but was later featured in the print version of the Journal in 2020. At the time of the full-text screening process, the print version of the article was not available.
In regards to the overall complication and 90-day mortality rate, please refer to the Methods section, where we explain how we extracted data from the studies with overlapping populations, to ensure completeness of data:
“Between studies with overlapping populations, we included those having the largest number of patients or reporting granular data on the outcomes of interest. In one occasion, analyses on additional outcomes were presented in two eligible articles; hence, data were extracted from both but their population was not summed in the overall cohort numbers, as they constituted additional analyses on the same subjects”
2. All values of continuous outcomes in Sarkaria et al.’s paper are expressed as median (range) and therefore, have to be converted to mean ± SD, in order to be incorporated into a meta-analysis.
Please refer to our Methods section, were we explain how we used a previously validated method to convert values expressed in median (range) to mean ± SD:
“When continuous data were presented as medians and ranges or interquartile ranges, we estimated the respective means and SDs by applying the methods described by Hozo et al. and Wan et al.[28,29]”.
The only other study that specifically refers to anastomotic leak grade is van der Sluis et al., in which case grade I anastomotic leaks were 0 in both groups and did not affect outcomes. We are aware of this issue and we clearly state it in the Limitations section:
“Similar concerns about the heterogeneity of our sample arise due to varying definitions and assessment protocols for complications…”
3. Similarly, the study by Jeong et al. expresses continuous variables as medians and interquartile ranges. We therefore used the aforementioned validated method by Wan et al. to convert them into means and standard deviations.
4. We have removed the transhiatal group from our analysis, according to your suggestion. We performed all meta-analyses again and have updated our text, as well as all of our tables, figures and supplementary material to reflect these changes. No statistically significant change was observed in any of the outcomes.
It is clearly mentioned in the Methods section that the mortality outcome in the study by Meredith et al. refers to 90-day mortality:
“Secondary end-points included R0 resections, lymph node harvest, and peri-operative adverse events (AE) (< 90 days following surgery), including pneumonia, cardiac arrhythmia, deep vein thrombosis (DVT)/pulmonary embolism (PE), wound infection, anastomotic leak or stricture, as well as 90-day mortality”
Please refer to the previous comments in regards to the conversion of median range/IQR to mean ± SD.
5. Operation and blood loss are both expressed as median (range). Please refer to the previous comments in regards to the conversion of median range/IQR to mean ± SD.
6. As evidenced in Table 1, although both studies by Sugawara et al. and Mori et al. were performed in the same center, the study periods are fully non-overlapping, thus their study populations are entirely separate.
We are aware of the important distinction between the transhiatal and the conventional transthoracic approach; however, our goal was to synthesize all current data on the use of robotics in esophagectomy regardless of the approach (transthoracic or transhiatal or transhiatal combined with cervical), given that outcomes between transhiatal and transthoracic approaches have been shown to be mostly similar. Having said that, we do understand the limitations that this approach imposes on the conclusions of this study. For this reason, we clearly mention this fact as a major limitation of our study in the limitations section:
“First, we included patients undergoing esophagectomy with either the transthoracic or the transhiatal approach. Although both approaches lead to similar outcomes, the transhiatal approach offers a limited capacity for an extended lymphadenectomy [47].”
Unfortunately, the limited number of studies utilizing the robotic transhiatal approach precludes any meaningful subgroup analysis that would enable more robust conclusions to be drawn.
7. We included all studies that met our inclusion criteria, as clearly defined in the Methods section. Notably, we did not impose any sample size limitations, as this could be a source of bias. Therefore, the study by Diez del Val et al. should be included in the analysis. The results of that study are weighted appropriately in the statistical analysis in order to reflect the smaller sample size compared to the other studies.
Please refer to for further reading:
https://doi.org/10.1371/journal.pone.0059202, https://skin.cochrane.org/sites/skin.cochrane.org/files/public/uploads/CSG-COUSIN_March%202015_M%20Grainge.pdf
Reviewer 3 Report
Dear Sirs,
I congratulate the authors on their extensive research in this field and putting together this very interessting and important article. RAMIE is important and will gain more interested in the future.
However, I have some concerns to raise:
- Line 32/34: Same sentences repeated. Please condensate the abstract a bit.
- I was wondering if it was possible if it was possible to extract the different types of procedures from the different studies and mention them accordingly.
- how do the authors explain the different proportions of SCC in both cohorts even though most tumors were located in the distal 1/3 in both cohorts?
- please consider using the abbreviation "OE" for open esophagectomy.
- line 256: add an "n" to comparison
- line 257: ad an "A" to RAMIE
- line 349: Should read 3.5.9: Chylothorax
My major concern is regarding the discussion: I feel it reads not very consistantly, not proving much new information to the reader. I would recommend restructuring the discucssion , making it a bit more interesting and structured. Additionally, important new aspects, which may not be extracted of the studies, but which are essential to be kept in mind, i.e. postoperative quality of health, postoperative function etc. need to mentioned at least in the discussion (read PMID 33122987 for example).
.
I am looking forward to reading the improved paper.
Kind regards
Author Response
The authors would like to thank the reviewer for their comments.
- We have condensed the abstract to avoid repeating sentences, according to your suggestion.
- We now list the different types of procedures in Table 1 according your suggestion.
- The proportions of SCC in both cohorts are comparable (72.7% vs 74.4%), but adenocarcinoma and SCC percentages were juxtaposed in the table by accident. We have amended the mistake so that all proportions now appear correctly.
- We now use the abbreviation “OE” instead of open esophagectomy throughout the manuscript according to your suggestion.
- Line 256: We have corrected the mistake.
- Line 257: We have corrected the mistake.
- Line 349: We have corrected the mistake, as also appropriately pointed out by Reviewer 1.
- Our discussion follows a format and structure similar to that of our results, with each paragraph addressing a major outcome category (e.g., overall complication rate, pulmonary complication rate, oncological outcomes, etc.). In each paragraph, we make an attempt to provide an explanation for our findings based on hypotheses supported by prior evidence in the literature rather than simply reiterating the results; thus, we hope that we are able to provide interesting and valuable information to the readers that enhances their understanding of the topic. The nature of our study is limited to synthesizing data from previously published studies and for this reason, we are not able to safely draw further conclusions or provide more new information than what is already provided in this manuscript.
However, we agree that outcomes that could not be synthesized due to certain methodological limitations (e.g., use of different time frames or measurement scales) but are still useful in clinical practice, such as quality of life, deserve to be mentioned in the discussion. Thus, we added a small segment in our discussion in regards to these outcomes, according to your suggestion.
Round 2
Reviewer 2 Report
The authors reviewed 10 articles that compared RAMIE with OE and found 674 patients in the RAMIE group and 1303 patients in the OE group. In terms of early outcomes, they concluded that the RAMIE has several benefits over the OE (length of stay and pulmonary complications). This is a well-written work, and the authors reviewed recently published RAMIE papers thoroughly. I have a few inquiries.
- In Sarkaria's study, the RAMIE group spent 13 days in the ICU and the OE group spent 14 days. In comparison to other research, I believe the ICU stay was excessive.
- Some centers used total operation time while others used console time when calculating operative time. The authors should double-check the operation time, in my opinion.
- As in prior studies, the clinical stages of the RAMIE group were better than the OE groups. This finding should be carefully labeled by the authors. Higher stages are related to more postoperative complications and a lower long-term survival rate.
Author Response
“The authors reviewed 10 articles that compared RAMIE with OE and found 674 patients in the RAMIE group and 1303 patients in the OE group. In terms of early outcomes, they concluded that the RAMIE has several benefits over the OE (length of stay and pulmonary complications). This is a well-written work, and the authors reviewed recently published RAMIE papers thoroughly. I have a few inquiries.”
Response: Thank you for your comments.
“In Sarkaria's study, the RAMIE group spent 13 days in the ICU and the OE group spent 14 days. In comparison to other research, I believe the ICU stay was excessive.”
Response: Thank you for your observation. This is probably the result of outliers that skew the mean ICU length of stay towards higher numbers compared to other studies. Notably, the median ICU length of stay for the RAMIE group in Sarkaria’s study is only 8 days, but the mean increases to 13 as a result of an outlier evident by the upper range which is 34 days. It is also important to note that the ICU admission rate in Sarkaria’s study is 20% in the OE group and 8% in the RAMIE group, and is thus representative of only a small subset of the overall population. This contrasts with the remaining studies, in which all patients were admitted to the ICU following esophagectomy for overnight observation. Given the above, we have decided to exclude Sarkaria’s data for the outcome of ICU length of stay, as they introduce significant heterogeneity to the rest of the data and only represent patients with surgical complications that required prolonged ICU admission or readmission.
“Some centers used total operation time while others used console time when calculating operative time. The authors should double-check the operation time, in my opinion.”
Response: We have double-checked all the definitions for operating time provided in the original papers. All studies included in the meta-analysis for this outcome define operative time as the time from first incision to closure. Only the study by Diez del Val et al. provides a different definition for the operative time but no data are provided for this outcome. Significant differences in the operative time between different authors may be attributed to their experience with the technique as well as the different type of RAMIE approach used (transhiatal vs. transthoracic).
“As in prior studies, the clinical stages of the RAMIE group were better than the OE groups. This finding should be carefully labeled by the authors. Higher stages are related to more postoperative complications and a lower long-term survival rate.”
Response: Thank you for your comment. We have included your remark in the limitations of our study.
Reviewer 3 Report
-
Dear Sirs,
I thank the authors providing the opportunity to read the revised paper and I see the effort that has been made.
- However, I have some mayor concerns which have not been solved by the revised form:
The authors define „RAMIE“: Only robotic throacic approaches, no matter what the abdominal approach was. I consider this to be problematic, if not wrong and disagree with the complete study design. Therefore, the paper should not be accepted in its current form.
- Further questions rose while rereading it:
- Why is there pulmonary complications and then again pneumonia and ards? why is mentioned twice in the outcomes section as primary and secondary outcome?
- Sugawara et al, Mori et al. and partly Diaz de Vale als include transhiatal apporaches in their studies: According to my understanding, this does not meet the inclusion criteria mentioned above?
- Weksler not availabel => does he really meet the inclusion criteria
- Minor revisions:
- numbers in table 2 for g3
- sentence in the abstract still twice
- new heading in table 1
- figure 1 needs to be further up
Therefore, I unfortunatelly may not recommend the acceptance and publication of this paper.
Author Response
“Dear Sirs,
I thank the authors providing the opportunity to read the revised paper and I see the effort that has been made.
However, I have some mayor concerns which have not been solved by the revised form:
The authors define „RAMIE“: Only robotic throacic approaches, no matter what the abdominal approach was. I consider this to be problematic, if not wrong and disagree with the complete study design. Therefore, the paper should not be accepted in its current form.”
Response: Thank you for your input. We further classified included procedures as totally robotic (TRAMIE), in cases where the robotic approach was utilized for both the thoracic and the abdominal part, and hybrid if the robotic approach was utilized only for the thoracic part of the operation. New metanalyses, with separate results for each subgroup, are presented in the Results section.
"Further questions rose while rereading it:
Why is there pulmonary complications and then again pneumonia and ards? why is mentioned twice in the outcomes section as primary and secondary outcome?”
Response: The primary outcome is overall pulmonary complications which encompasses all pulmonary complications, including both pneumonia and ARDS, as well as other complications such as pleural effusion, atelectasis, and respiratory failure. Pneumonia and ARDS were also analyzed separately as secondary outcomes, being the two most major pulmonary complications following esophagectomy. Given our results, it appears that the decreased pneumonia rates are primarily responsible for the decreased overall pulmonary complication rate associated with RAMIE.
“Sugawara et al, Mori et al. and partly Diaz de Vale als include transhiatal apporaches in their studies: According to my understanding, this does not meet the inclusion criteria mentioned above?”
Response: As we clarify in the Methods section: “studies were included independently of the surgical approach (i.e., transthoracic [Ivor-Lewis or McKeown], or transhiatal)”. Even though transthoracic and transhiatal representing different surgical approaches, results from prior studies have shown that they lead to similar outcomes. All transhiatal procedures included in this meta-analysis include a video-assisted cervical approach to perform mediastinal lymph node resection that emulates (albeit imperfectly) the transthoracic approach. Our primary goal was to synthesize all current data on the use of robotics in esophagectomy regardless of the approach.
“Weksler not availabel => does he really meet the inclusion criteria”
Response: The study from Weksler et al. represents an NCDB database analysis. It does meet our inclusion criteria given the definitions provided, however, we do not include it in the quantitative synthesis part of our study, as its population would overlap with the populations of single-center studies. It is only included in the qualitative part of the study for comparison purposes.
“Minor revisions:
numbers in table 2 for g3
sentence in the abstract still twice
new heading in table 1
figure 1 needs to be further up”
Response: We have corrected the numbers in Table 2 and deleted the duplicate sentence in the abstract.